# Implication of p16 Promoter Methylation, the BRAFV600E Mutation, and ETS1 Expression Determination on Papillary Thyroid Carcinoma Prognosis and High-Risk Patients’ Selection

**DOI:** 10.3390/biomedicines13071583

**Published:** 2025-06-27

**Authors:** Stefana Stojanović Novković, Sonja Šelemetjev, Milena Krajnović, Ana Božović, Bojana Kožik, Uršula Prosenc Zmrzljak, Tijana Išić Denčić

**Affiliations:** 1Department of Endocrinology and Radioimmunology, Institute for the Application of Nuclear Energy—INEP, University of Belgrade, Banatska 31b, Zemun, 11080 Belgrade, Serbia; stefana@inep.co.rs (S.S.N.); sonja@inep.co.rs (S.Š.); 2Laboratory for Radiobiology and Molecular Genetics, Vinča Institute of Nuclear Sciences, National Institute of the Republic of Serbia, University of Belgrade, Mike Petrovića Alasa 12–14, Vinča, 11351 Belgrade, Serbia; mdragic@vin.bg.ac.rs (M.K.); anabozovic@vin.bg.ac.rs (A.B.); bojana86@vin.bg.ac.rs (B.K.); 3Molecular Biology Laboratory, BIA Separations CRO—Labena d.o.o, 1000 Ljubljana, Slovenia; ursula.prosenc@biaseparationscro.com

**Keywords:** tumor marker, prediction, tumor aggressiveness, epigenetic, PTC, patients’ stratification

## Abstract

**Background/Objectives**: Papillary thyroid carcinoma (PTC) is the most common malignancy of the endocrine system, characterized by various molecular alterations. This study evaluates the relationship between p16 promoter methylation status, BRAFV600E mutation presence, and ETS1 (E26 transformation-specific) expression, aiming to better understand their clinical significance and to enhance the risk stratification of PTC patients. **Methods**: p16 promoter methylation was analyzed by methylation-specific PCR (MSP), BRAFV600E by mutant allele-specific PCR amplification (MASA), ETS1 mRNA expression by quantitative PCR (qPCR), ETS1 protein expression by immunohistochemistry (IHC), and Western blot. All tested factors were further associated with the occurrence of unfavorable clinicopathological data of the patients. **Results**: While p16 methylation did not correlate with adverse clinical parameters or BRAFV600E mutation presence, it was significantly associated with the increased ETS1 mRNA levels. Combined p16 methylation with high ETS1 protein levels was significantly associated with advanced pT and pTNM stages. BRAFV600E-mutated PTC cases with p16 methylation showed increased mRNA and protein ETS1 expression. **Conclusions**: Therefore, although p16 methylation could not be used as a standalone prognostic marker, its association with elevated ETS1 levels points to its potential involvement in tumor progression and adverse clinical outcomes, particularly in BRAFV600E-mutated PTCs. Deeper insights into these interactions may enhance PTC prognosis and the selection of high-risk patients.

## 1. Introduction

Papillary thyroid carcinoma (PTC), which accounts for 80–85% of all thyroid cancers, has distinct clinical and molecular features [1,2,3]. Despite its generally favorable prognosis, a subset of PTC patients develops recurrence and metastasis, underscoring the need for refined diagnostic and prognostic tools to guide personalized treatment strategies [4,5,6]. PTC is characterized by a wide variety of cytomorphological and pathological features, caused by a complex interaction of genetic and epigenetic changes that may drive its onset and progression [1,3,7,8]. This complex molecular landscape involves genetic mutations, epigenetic modifications, and altered gene expression patterns, each contributing to the heterogeneity observed in clinical behavior [1,2,7,8,9].

Epigenetic modifications, such as DNA methylation, play a pivotal role in regulating gene expression. Promoter methylation of tumor suppressor genes leads to their inactivation, disrupting cell cycle control and promoting uncontrolled proliferation, and has been observed in various cancers, including PTC [9,10,11]. One significant epigenetic alteration occurring in PTC is the hypermethylation of the *p16* promoter, which has been demonstrated to contribute to tumor progression by inactivating the *p16* tumor suppressor pathway [12,13,14]. Studies indicate that *p16* promoter methylation can lead to the silencing of genes that would otherwise inhibit tumor progression, facilitating a more aggressive phenotype in PTC patients [14]. Moreover, promoter hypermethylation is considered an early event in thyroid carcinogenesis, facilitating the transition of precancerous states to invasive carcinoma [13].

In addition to p16 methylation, the *BRAFV600E* mutation has emerged as a prevalent genetic alteration in PTC, being detected in approximately 40–75% of cases [2,15]. This mutation not only plays a crucial role in enhancing tumor aggressiveness [16], but also interacts with other epigenetic modifications, such as methylation, potentially amplifying the oncogenic pathways involved in the progression of PTC [17]. The mutation is usually absent in other types of differentiated thyroid carcinomas, underscoring its specificity to PTC and its diagnostic relevance [7,17,18]. The interplay between the *BRAFV600E* mutation and *p16* methylation might not only highlight the intricate mechanisms behind the PTC development but also elucidate potential therapeutic targets for managing this malignancy [19].

Furthermore, the expression levels of *ETS1*, both at the mRNA and protein levels, have emerged as significant factors in PTC development and progression [20,21]. ETS1 is a transcription factor related to cell differentiation and proliferation [22,23]. Therefore, it has been implicated in various processes commonly associated with tumor expansion, such as tumor growth, invasion and angiogenesis [24,25]. Research has demonstrated that altered ETS1 expressions may be indicative of PTC aggressiveness and may correlate with various clinicopathological features, such as lymph node metastasis and disease recurrence [26,27]. It has been shown that ETS1 is essential for the *BRAFV600E*-induced signaling pathway that promotes tumor survival and progression of PTC [28]. Furthermore, in human ovarian cancer ETS1 regulates its target genes mainly by DNA methylation [29].

Therefore, the interplay between *p16* promoter methylation, the *BRAFV600E* mutation, and *ETS1* expression may represent a significant aspect of PTC biology. By understanding these molecular alterations and their patterns, we could enhance our comprehension of PTC’s aggressive nature development and potentially contribute to the improvement of prognostic or therapeutic strategies in managing this prevalent malignancy.

## 2. Materials and Methods

### 2.1. Patients and Tissue Samples

The patients with PTC underwent surgical treatment at the Clinic for Endocrine Surgery, Clinical Center of Serbia, Belgrade. The resected tissue sample from each patient was split into two pieces: one half was immediately frozen in liquid nitrogen and stored at −80 °C for further use, while the other was fixed in formalin and embedded in paraffin (FFPE sample preparation) [30]. All participants provided informed consent. The study was conducted in accordance with the Declaration of Helsinki and was approved by the Ethics Committee of the Faculty of Medicine, University Clinical Center of Serbia.

A pathologist made a histopathological diagnosis based on recognized cytohistopathological criteria [31]. Final diagnosis and clinical information on patient age and sex, tumor dimensions, its dissemination through the gland (intraglandular dissemination, ID), outside of the gland (extrathyroidal invasion, Ei), local (lymph node metastases, Lnm), and distant metastases were obtained by reviewing pathology reports. Carcinomas were graded according to pathologic tumor-node-metastasis (pTNM) stage [32]. All cases were subsequently classified according to the degree of tumor infiltration (DTI) [33]: 1—fully encapsulated tumors; 2—non-encapsulated tumors with no invasion of thyroid capsule; 3—tumors with invasion of thyroid capsule; and 4—tumors with Ei. 

The total number of PTC patients included in this study was 57, of which 41 were females (71.9%) and 16 were males (28.1%). The patients’ ages ranged from 14 to 89 years at diagnosis (53.0 ± 16.8 years, mean ± SD). Tumor sizes ranged from 6 to 120 mm (32.5 ± 18.5 mm, mean ± SD). ID was observed in 32 (56.1%), Lnm in 11 (19.3%) and Ei in 17 cases (29.8%). In our PTC sample series, no patient had a known distant metastasis.

### 2.2. RNA and DNA Isolation

The isolation of nucleic acids was performed as explained previously [30]. Shortly, TRIzol reagent (Invitrogen, Carlsbad, CA, USA) was used for RNA extraction, while DNA was extracted by Proteinase K (Sigma-Aldrich, Taufkirchen, Germany). The concentration and quality of the isolated RNA/DNA were determined with an Epoch microplate spectrophotometer (BioTek, Winooski, VT, USA).

### 2.3. p16 Methylation Analysis

Methylation-specific PCR (MSP) was used to assess DNA methylation patterns in the *p16* promoter CpG islands. The EZ DNA Methylation-Lightning™ kit (Zymo Research, Orange, CA, USA) was used to convert genomic DNA (100–500 ng) to sodium bisulfite, following the manufacturer’s instructions. To perform MSP reactions, 1 μL of bisulfite-modified DNA was utilized per 10 μL. In a final volume of 25 μL, the PCR mixture included 1× PCR buffer (16 mmol/L ammonium sulfate, 67 mmol/L Tris-HCL, pH 8.8, 10 mmol/L 2-mercaptoethanol), 6.7 mmol/L MgCl_2_, dNTP (each at 1.25 mmol/L), primers (300 ng each per reaction), 5% DMSO, and 0.4 mg/mL bovine serum albumin (BSA). Reactions were hot-started at 95 °C for 5 min before adding 1 unit of Taq polymerase (Thermo Fisher Scientific, Rockford, IL, USA). Amplification was performed in an Applied Biosystems 2720 temperature cycler (Applied Biosystems, Foster City, CA, USA) for 40 cycles (45 s at 95 °C, 45 s at the annealing temperature unique to each primer set, and 60 s at 72 °C, followed by a final extension of 4 min at 72 °C). The primer sequences for each reaction are reported in Table 1. DNA from a healthy donor’s peripheral blood lymphocytes served as a negative control for the methylation alleles. The same leukocyte DNA was methylated in vitro with excess SssI methyltransferase (New England Biolabs, Ipswich, MA, USA) to generate completely methylated DNA at all CpG sites and used as a positive control for all genes. PCR products were separated by electrophoresis on 6% acrylamide gels, stained with silver nitrate and sodium carbonate.

### 2.4. The BRAFV600E Mutational Analysis

To detect the presence of the *BRAFV600E* point mutation we performed mutant allele-specific PCR amplification (MASA), as previously described [34]. The sequences of the primers utilized (two forward primers, one for wild type and one for the *BRAFV600E* mutant, and one reverse primer) are shown in Table 1.

### 2.5. Reverse Transcription and Quantitative PCR of ETS1 mRNA

To synthesize cDNA, 1 μg of extracted RNA was reverse transcribed using the iScript Select cDNA Synthesis Kit (Bio-Rad Laboratories, Inc., Hercules, CA, USA) and random hexamer primers, following the manufacturer’s instructions and as previously described [26]. The level of *ETS1* mRNA expression was normalized to the *GAPDH* level.

Table 1 lists the primer sequences that were employed.

### 2.6. Quantification of ETS1 Protein Levels

The level of ETS1 protein expression was determined immunohistochemically (IHC) in 54 PTC cases and by Western blot in 30 PTC cases. The anti-ETS1 monoclonal antibody (clone JM92-32, Cat #MA5-32732, Thermo Fisher Scientific, Rockford, IL, USA) was applied in a 1:100 dilution for IHC and a 1:1000 dilution for Western blot. The IHC score was gained by determining both the intensity of staining and the distribution of staining (the percentage of ETS1-positive cells), as previously described in detail [26]. In Western blot analysis, the level of ETS1 expression in PTC was normalized to the level of β-actin as an endogenous control (ETS1 relative expression). The anti-β-actin antibody (clone AC-15, Cat #MA1-91399, RRID: AB_2273656, Thermo Fisher Scientific, Rockford, IL, USA) was applied in a 1:2000 dilution for Western blot. In addition, ETS1 expression was also determined in the matched nonmalignant thyroid tissue (NMT) resected from the opposite lobe of the patient. The fold change of ETS1 was calculated as the relationship of ETS1 relative expression in PTC vs. ETS1 relative expression in matched NMT. All values of ETS1 fold change greater than 1 indicate cases with upregulated ETS1 expression, which means that the levels of ETS1 are higher in malignant tissue compared to matched NMT. The protocols were previously explained in detail [26].

### 2.7. Statistical Analysis

All variables determined in this study were first tested for the distribution type by the Shapiro-Wilk test. None of them followed normal distributions; therefore, the non-parametric tests were applied for further analysis. Spearman’s correlation was applied for testing the correlation of p16 methylation, BRAFV600E occurrence, and ETS1 expression with the clinicopathological data of the patients. The χ^2^-test was applied for testing the associations of tested parameters with the occurrence of unfavorable clinical parameters of PTC patients, as well as to test the group differences through the examined categories. The Mann-Whitney U test and median test were applied for testing the difference between two groups of samples. The results were statistically significant at *p* < 0.05. Statistical analysis was carried out using SPSS software (SPSS 16.0, Chicago, IL, USA).

## 3. Results

### 3.1. The p16 Promoter Methylation

There were 57 PTC patients included in this study, of which 57.9% (33/57) samples had the *p16* promoter methylated. The *p16* promoter methylation had no statistically significant association (all *p* > 0.05, χ^2^-test, Appendix A) with any of the tested unfavorable clinical parameters in PTC patients.

### 3.2. The p16 Promoter Methylation and the BRAFV600E Mutation Status

From the 57 PTC cases tested in this study, 70.2% (40/57) had the *BRAFV600E* mutation and 38.6% (22/57) had mutually methylated *p16* promoter and the *BRAFV600E* mutation present (Table 2). There was no significant association of the p16 promoter methylation and the *BRAFV600E* mutation (*p* = 0.497, tested by χ^2^-test) in the tested PTC cohort.

### 3.3. The p16 Promoter Methylation and ETS1

The *p16* promoter methylation was further associated with the *ETS1* mRNA levels, relative ETS1 protein levels and ETS1 protein fold change (defined as the ratio of the ETS1 protein levels in PTC and matched nonmalignant thyroid tissue, NMT). All variables were tested for the distribution type and none showed normal distribution. Therefore, a non-parametric set of tests was further applied. Figure 1 summarizes the number of PTC patients included in each analysis. *P16* methylation and *BRAFV600E* were determined in all tested cases. mRNA expression of *ETS1* was determined in 41/57 PTC cases included in the study due to the sample availability and the poor quality of the mRNA extraction. ETS1 protein expression was determined in 54/57 PTC cases by IHC, as in 3 cases, due to the small size of the tumor (microcarcinoma), the PTC tissue was “spent” for FFPE slides to obtain a definitive diagnosis. Finally, ETS1 was determined by Western blot in 30/57 PTC cases as in 30 cases the matched nonmalignant thyroid tissue (NMT) resected from the opposite lobe of the patient was available.

The methylation status of the *p16* promoter correlated significantly with the level of *ETS1* mRNA (r = 378, *p* = 0.015, Spearman’s test of correlation) in the tested PTC. Furthermore, the distribution (*p* = 0.017, tested by Mann-Whitney U test) and medians (*p* = 0.038, tested by Median test) of *ETS1* mRNA levels differed significantly across the *p16* methylation categories in the tested PTC cohort. Therefore, the level of *ETS1* mRNA is significantly higher in PTCs with methylated *p16* promoter than in the *p16* nonmethylated group (Figure 2A).

Although there was a noticeable increase in the relative levels of ETS1 protein expression (Figure 2B), as well as in ETS1 fold change (Figure 2C), in PTCs with methylated *p16* vs. PTC cases with unmethylated *p16*, the difference did not achieve statistical significance (*p* > 0.05, tested by Mann-Whitney U test). Western blot analysis of ETS1 protein expression in PTC tissue and adjacent NMT is shown in Figure 3.

The mutual distribution of *p16* methylation and the level of ETS1 protein expression were additionally determined by immunohistochemistry in 54 of 57 PTC cases included in the study. The tested group was then split into two subgroups according to the median value of ETS1 protein levels in the total PTC sample: high expressing subgroup included samples with ETS1 protein levels equal or higher than median ETS1 value in total sample and low expressing group included PTC samples with ETS1 protein levels lower than the same median value. The representative micrographs of IHC staining of ETS1 in PTC are shown in Figure 4. The results were again associated with the p16 methylation status and the unfavorable clinical parameters of the patients (Table 3). The combination of p16 methylation and the high levels of ETS1 protein was significantly associated with the pT and pTNM stage of the PTC patients (*p* < 0.05, tested by χ^2^-test).

### 3.4. The BRAFV600E Mutation Status and ETS1

There was no significant difference in the distribution or medians of ETS1 levels (mRNA or protein) across categories which included the *BRAFV600E* presence (Figure 5). In particular, the distribution of mRNA *ETS1* levels was the same in *BRAFV600E* mutated and unmutated PTC cases (*p* = 0.151, tested by Mann-Whitney U test), as well as were the medians (*p* = 0.724, tested by the Median test). Although there was an increase in the relative levels of ETS1 protein expression (Figure 5B) and ETS1 fold change (Figure 5C) in PTCs with *BRAFV600E* compared to PTC cases without the mutation, the difference was not statistically significant (all *p* > 0.05, Mann-Whitney U test and Median test).

### 3.5. The Combination of the p16 Promoter Methylation, the BRAFV600E Mutation and ETS1 Levels

There was a significant difference in the distribution of ETS1 protein levels across categories which included the combination of *p16* methylation and the *BRAFV600E* presence (*p* = 0.010, tested by χ^2^-test, Table 4). The *BRAFV600E* mutated PTC cases with methylated *p16* promoter correlated significantly to the higher *ETS1* mRNA expression (*p* = 0.003, r = 0.456, Sperman’s test of correlation), higher ETS1 protein expression (*p* = 0.005, r = 0.775, Sperman’s test of correlation), as well as to the ETS1 protein upregulation (*p* = 0.021, r = 0.609, Sperman’s test of correlation). On the other hand, the mutual presence of methylation in the *p16* promoter with the *BRAFV600E* mutation and high ETS1 protein expression in PTC did not associate with the higher pT (0.067, tested by χ^2^-test) and pTNM stage (0.129, tested by χ^2^-test).

## 4. Discussion

The present study investigates the interplay between *p16* promoter methylation status, the *BRAFV600E* mutation, and *ETS1* gene and protein expression in patients with papillary thyroid carcinoma (PTC). Our findings shed light on the complex interactions that may exist within this malignancy. While *p16* promoter methylation does not correlate significantly with unfavorable clinical parameters or *BRAFV600E* mutation status, it is associated with increased *ETS1* mRNA expression in PTC patients. This intriguing observation may have implications for understanding the tumor behavior.

According to our results, the methylation status of the *p16* promoter does not correlate with adverse clinical parameters typically associated with PTC. This lack of correlation implies that *p16* methylation alone may not serve as a biomarker for predicting aggressiveness or metastatic potential of PTC. This finding isn’t consistent to the previous findings in PTC, as Wang et al. (2013) [14] showed that *p16* methylation is associated with thyroid cancer metastasis and AMES (Age, Metastases, Extent and Size) stratification of PTC patients. Therefore, while *p16* is an important tumor suppressor gene whose methylation status could be used as a diagnostic marker of PTC [12,13], its methylation status may not be a stand-alone factor critical for PTC progression and may not have a straightforward relationship with the clinical outcomes in patients with PTC.

Moreover, we found no correlation between *p16* promoter methylation and *BRAFV600E* mutation in our PTC cohort. As long as *BRAFV600E* is prevalent in aggressive forms of PTC and contributes to disease progression [15], the association between *BRAFV600E* and methylation alterations is not consistently confirmed [17,35,36]. This distinction may indicate that *p16* methylation could represent an epigenetic alteration that occurs without directly influencing the oncogenic activity of *BRAFV600E* and that maybe *p16* methylation operates through alternate pathways that do not significantly overlap with the oncogenic effects of *BRAFV600E* mutations in PTC. In other words, this lack of correlation reinforces the notion that these two alterations may operate via distinct pathways in PTC. Previous research also failed to establish a correlation between *p16* methylation and specific oncogenic mutations in thyroid cancers, indicating that *p16* methylation may represent an event that occurs independently of other genetic alterations commonly explored in PTC [12,13]. This independence raises questions about the pathways involved in PTC development, suggesting a more complex interplay between genetic mutations and epigenetic modifications in thyroid tumors. It is also possible that maybe methylation of other tumor suppressor genes or alternative pathways contribute more significantly to the clinical outcomes in PTC than *p16* status alone. This is noteworthy, particularly as some studies have found correlations between BRAF mutations and increased methylation in other carcinoma types [17,19,35,36].

In contrast, our results show that *ETS1* mRNA levels were significantly higher in PTCs with a methylated *p16* promoter than in those without, which presents an intriguing dimension to the role of ETS1 in thyroid carcinogenesis. Our findings suggest that *p16* epigenetic silencing may result in enhanced transcriptional activity of *ETS1*, which is consistent with prior research indicating ETS1 as a modulator of carcinogenic processes [24,25,29]. However, it is crucial to note that despite the apparent rise in *ETS1* expression at the mRNA level, there was no statistically significant change in the corresponding protein expression levels in this investigation. Namely, while fold changes in ETS1 protein levels were noted between the groups, the statistical significance was not achieved in this study. This implies a nuanced role for ETS1 in PTC where mRNA elevation does not necessarily translate to protein abundance with immediate clinical relevance [20,26]. Although this discrepancy might be attributed to the small sample size or variability in protein expression, which can often result in difficulties achieving clear statistical distinctions, it might also reflect post-transcriptional regulation, a concept well known in cellular biology and frequently associated with cancer development [37,38,39]. It assumes that mRNA levels do not always translate linearly to protein levels. Therefore, while transcriptional upregulation of *ETS1* occurs, there could exist additional regulatory mechanisms, such as post-transcriptional silencing by microRNAs, post-translational modifications, protein stability or interactions with other signaling molecules, which influence ETS1 protein translation rate and warrant further investigation [37,38,39].

ETS1 often promotes oncogenic pathways by interacting with key transcription factors and signal transduction pathways. It is a downstream target of Ras/Raf/MEK signaling, which is frequently activated in PTC due to the *BRAFV600E* mutation [40,41]. Our observation that elevated *ETS1* expression is particularly pronounced in the presence of the *BRAFV600E* mutation supports the hypothesis that these two factors may synergistically enhance tumor progression by over-activating downstream signaling pathways linked to cellular proliferation and migration. Furthermore, it was shown that *ETS1* can be activated by MAPK-mediated phosphorylation [28,29,42].

An essential finding of our study is the significant association between the combination of *p16* methylation and high levels of ETS1 protein with advanced pT and pTNM stages of PTC. This emphasizes the importance of understanding how these molecular alterations collectively impact tumor progression. While *p16* methylation alone may not directly influence clinical outcomes, its interaction with ETS1 could still play a role in tumor advancement and may contribute to more aggressive tumor behavior. This highlights the potential utility of assessing both *p16* methylation and ETS1 expression in predicting clinical outcomes and affirms their potential as indicators of PTC severity. Given the evolving nature of PTC research, these markers may serve as a composite biomarker to guide prognosis, especially in patients at increased risk for aggressive disease. This perspective aligns with the evolving paradigm, focusing not only on genetic mutations but also on epigenetic modifications and their effects on tumor behavior [9,11,12,13,14,43]. Moreover, we observed that within the *BRAFV600E*-mutated subset of PTC cases, methylated *p16* was correlated with heightened *ETS1* mRNA expression, increased ETS1 protein expression, and protein upregulation. Furthermore, although insignificant, some tendencies could be noticed in PTC cases with all three tested parameters positive (*p16* methylated, *BRAFV600E* mutated, and ETS1 protein levels high) to the higher pT and pTNM stages. This relationship suggests that co-targeting this pathway could be a novel approach in precision medicine for PTC patients displaying these specific molecular profiles. While this necessitates further exploration into the functional roles of these molecules within the tumor development, the correlation of *BRAFV600E*-mutated PTC cases with methylated *p16* and elevated ETS1 expression highlights a subtype of PTC that may benefit from more targeted therapeutic strategies. This supports the argument for stratifying PTC patients based on molecular profiles, which could ultimately guide individualized treatment regimens and predict response to therapy.

Nonetheless, our study has limitations that must be noted. The sample size, while adequate for initial explorations, may restrict the generalizability of our findings. Additionally, the focus on mRNA and protein expression without corresponding functional studies, gene silencing, or pathway inhibition experiments limits our capacity to draw definitive mechanistic conclusions about how *p16* methylation or the *BRAFV600E* mutation influences *ETS1* activity within the context of PTC. With all the limitations mentioned, most notably methodological limitations, a small sample size, and a lack of functional validation, the findings of this study should be regarded as a pilot study with substantial preliminary results that should be validated in further research. Future studies, possibly multicentric ones, incorporating larger cohorts and longitudinal follow-up (with recurrence and survival information), which would include a validation group of cases, as well as functional assays to dissect the roles of *BRAFV600E*, *ETS1*, and *p16* methylation more precisely, will be essential to validate our findings and clarify the underlying mechanisms. Although the translational relevance of our results is still limited, restricting the applicability of this marker panel in clinical decision-making, the study’s findings are encouraging. Therefore, in the future, with a larger sample size, cell culture experiments, an adequate in vivo model system, external validation or replication in an independent cohort, and a tested marker panel, this combination might happen to be of great help for the risk stratification of PTC patients and for identifying patients who may benefit from more aggressive therapeutic strategies.

## 5. Conclusions

In summary, our investigation underscores the complexity of molecular interactions in PTC and highlights the multifactorial nature of its pathogenesis, which includes methylation, mutations, and expression changes. While *p16* methylation could not be used as a standalone prognostic marker, its relationship with *BRAFV600E* and ETS1 could shed some light into PTC biology. Enhanced knowledge of these interactions may improve prognostic stratification of PTC patients and eventually adjust PTC patient management.

## Figures and Tables

**Figure 1 biomedicines-13-01583-f001:**
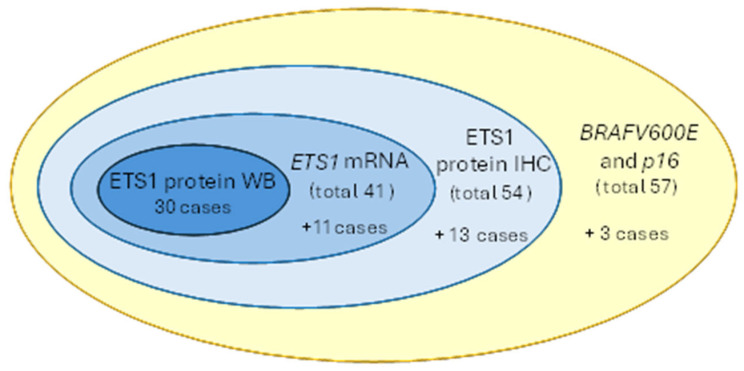
A diagram presenting the number of PTC cases included in each analysis. The numbers in the brackets represent the total number of PTC cases assessed for each analysis. The numbers presented below the total number of cases represent the additional number of PTC cases analyzed in each test, compared to the inner sample set (ellipsoid layer). The number of PTC cases with determined *p16* and *BRAFV600E* status is indicated in orange. ETS1 expression is shown in blue. PTC: papillary thyroid carcinoma, WB: Western blot, IHC: immunohistochemistry.

**Figure 2 biomedicines-13-01583-f002:**
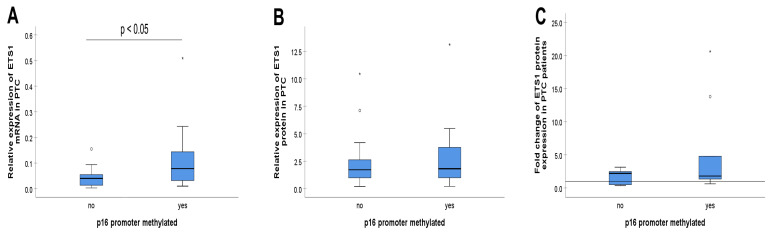
The relations between ETS1 expression and the *p16* promoter methylation in papillary thyroid carcinoma. (**A**) The relation between *ETS1* mRNA expression and the *p16* promoter methylation, (**B**) The relation of ETS1 protein expression and the *p16* promoter methylation, (**C**) The relation of ETS1 protein upregulation and the *p16* promoter methylation. The reference line presents value 1 for ETS1 protein fold change. *P16* promoter methylation was analyzed by methylation-specific PCR (MSP), relative *ETS1* mRNA expression was determined by quantitative PCR, relative ETS1 protein expression and its fold change were determined by Western blot. The measurement data are expressed via box plots. The boxes represent the median value (the black horizontal line inside the box) and the interquartile range (upper and lower value); the symbols ‘o’ represent extreme values, and the symbols ‘*’ represent outliers. For details, see Section 2.

**Figure 3 biomedicines-13-01583-f003:**
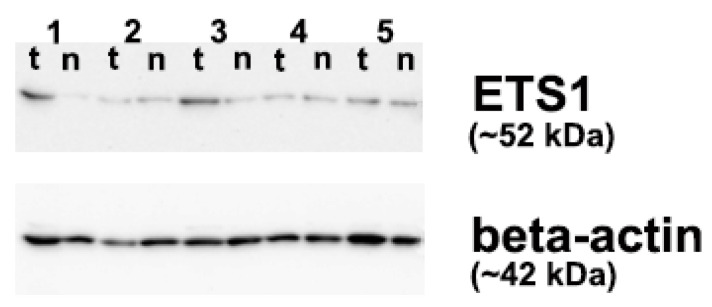
Western blot analysis of 5 papillary thyroid carcinoma (PTC) cases using anti-ETS1 antibody. 1–5: case numbers, t: tumor (PTC), n: matched non-malignant thyroid tissue. Cases 1–3 are the *BRAFV600E* positive PTCs.

**Figure 4 biomedicines-13-01583-f004:**
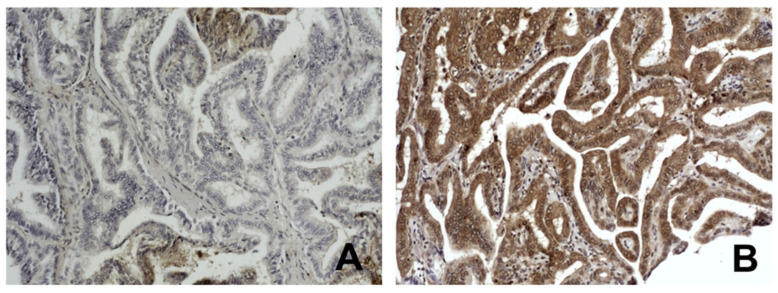
Immunohistochemical staining of ETS1 in papillary thyroid carcinoma (PTC): (**A**) Low expression of ETS1 in PTC case with pT2, pTNM I, and non-methylated p16 promoter, Original magnifications ×10, (**B**) High expression of ETS1 in PTC case with pT3 and pTNM III, and methylated p16 promoter. Original magnifications ×10. For details, see Section 2.

**Figure 5 biomedicines-13-01583-f005:**
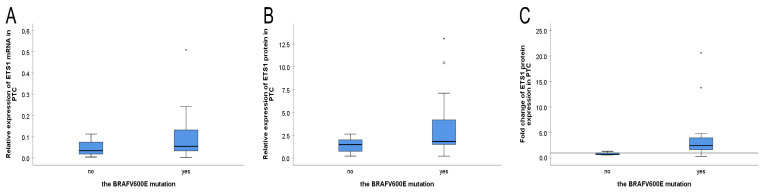
The relations between ETS1 expression and the *BRAFV600E* mutation in papillary thyroid carcinoma. (**A**) The relation between *ETS1* mRNA expression and *BRAFV600E*, (**B**) The relation of ETS1 protein expression and *BRAFV600E*, (**C**) The relation of ETS1 protein upregulation and *BRAFV600E*. The reference line presents value 1 for ETS1 protein fold change. The *BRAFV600E* mutation was analyzed by mutant allele-specific PCR amplification (MASA), relative *ETS1* mRNA expression was determined by quantitative PCR, relative ETS1 protein expression and its fold change were determined by Western blot. The measurement data are expressed via box plots. The boxes represent the median value (the black horizontal line inside the box) and the interquartile range (upper and lower value); the symbols ‘o’ represent extreme values, and the symbols ‘*’ represent outliers. For details, see Section 2.

**Table 1 biomedicines-13-01583-t001:** Sequences of primers.

Method	Primer	Sequence (5′-3′)
MASA	BRAF-f_wt (for wild type)	GTGATTTTGGTCTAGCTACAGT
	BRAF-f_mut (for BRAFV600E)	GTGATTTTGGTCTAGCTACAGA
	BRAF-r	GGCCAAAAATTTAATCAGTGGA
MSP	p16_U-f	TTATTAGAGGGTGGGGTGGATTGT
	p16_U-r	CAACCCCAAACCACAACCATAA
	p16_M-f	TTATTAGAGGGTGGGGCGGATCGC
	p16_M-r	GACCCCGAACCGCGACCGTAA
qPCR	ETS1-f	ACAGGACTCCATGGCAAACG
	ETS1-r	ATGAAGAAAGCCTGGTGCAGT
	GAPDH-f	GAAGGTGAAGGTCGGAGT
	GAPDH-r	GAAGATGGTGATGGGATTTC

PCR: Polymerase chain reaction, MASA: mutant allele-specific PCR amplification, MSP: methylation-specific PCR, qPCR: quantitative PCR. wt: wild type, mut: BRAFV600E mutant, f: forward primer, r: reverse primer, U: unmethylated (primer set for unmethylated modified DNA sequence), M: methylated (primer set for methylated modified DNA sequence).

**Table 2 biomedicines-13-01583-t002:** Association of the *p16* promoter methylation and the *BRAFV600E* mutation in PTC.

PTC		BRAFV600E		Total (n)
		No	Yes	
**p16 methylated**	**no**	6	18	24
	**yes**	11	22	33
**Total (n)**		17	40	57

PTC: Papillary thyroid carcinoma, n: number of cases.

**Table 3 biomedicines-13-01583-t003:** Association of *p16* methylation and the ETS1 protein levels with the occurrence of unfavorable clinicopathological parameters of PTC patients.

Clinicopathological Parameter		p16 Methylated and ETS1 Protein Levels High (n)		*p*-Value
		None or One	Both	
Gender	male	13	2	0.949
	female	29	10	
Intraglandular dissemination	no	18	5	0.941
	yes	24	7	
Lymph node metastasis	no	33	10	0.718
	yes	9	2	
Extrathyroid invasion	no	28	10	0.265
	yes	14	2	
Degree of tumor infiltration	1	9	6	0.251
	2	12	3	
	3	7	1	
	4	14	2	
pT	T1	8	1	**0.032**
	T2	14	9	
	T3	19	1	
	T4	1	1	
pTNM	I	15	2	**0.012**
	II	9	8	
	III	13	0	
	IV	5	2	

PTC: Papillary thyroid carcinoma, n: number of cases. *p*-value: statistical significance tested by χ^2^-test. Statistically significant results are bold.

**Table 4 biomedicines-13-01583-t004:** Association of the *p16* promoter methylation, the *BRAFV600E* mutation presence and the ETS1 protein expression in PTC.

PTC		The Level of ETS1 Protein Expression		Total (n)
		Low	High	
***p16* methylated/** ***BRAFV600E* mutated**	**None or one**	12	7	19
	**Both**	3	8	11
**Total (n)**		15	15	30

PTC: Papillary thyroid carcinoma, n: number of cases.

## Data Availability

The data presented in this study is available on request from the corresponding author.

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
