# Peer review of "Implication of p16 Promoter Methylation, the BRAFV600E Mutation, and ETS1 Expression Determination on Papillary Thyroid Carcinoma Prognosis and High-Risk Patients’ Selection"

_biomedicines, 2025, doi:10.3390/biomedicines13071583_

Round 1
Reviewer 1 Report
Comments and Suggestions for Authors
The study by Novkovic et al. investigates the relationship between p16 promoter methylation, BRAFV600E mutation, and ETS1 expression in papillary thyroid carcinoma (PTC). The authors report a significant correlation between p16 methylation and ETS1 mRNA levels. They also demonstrate that p16 methylation and elevated ETS1 expression are associated with more advanced pT and pTNM stages. Moreover, PTC patients harboring both BRAFV600E and p16 methylation exhibit increased ETS1 expression. These findings may help identify potential therapeutic targets and enhance prognostic evaluation in PTC.
The authors have adequately addressed my previous comments in the revised manuscript. I have only one remaining suggestion: it is redundant to perform both Spearman’s test and the Chi-square test for the same analysis (Supplementary Tables 1 and 2)
Author Response
Dear reviewer,
Thank you very much for your comments and highly graded valuation of our manuscript. We agree that presentation of both analyses might be redundant; therefore, in the revised version of our manuscript, the correlation analysis is omitted. Accordingly, we have removed the results of this analysis from section 3.1, deleted Supplementary Table 1, and renamed original Supplementary Table 2 to new Supplementary Table 1.
We sincerely hope that the corrections implemented will meet your requirements, that the manuscript is now sufficiently improved, and that it will lead to its approval.
Reviewer 2 Report
Comments and Suggestions for Authors
This manuscript investigates the prognostic relevance of p16 promoter methylation, BRAFV600E mutation, and ETS1 expression in papillary thyroid carcinoma (PTC). The study is addresses an important clinical question regarding molecular stratification in PTC. The concept of using molecular biomarkers for risk stratification in PTC is highly relevant and timely. However, it is not novel—biomarkers such as BRAFV600E, p16 promoter methylation, and ETS1 have already been extensively investigated in previous studies. While this study focuses on their combined assessment, its translational potential to clinical practice remains limited at this stage. While the findings are promising—particularly the correlation between p16 methylation, BRAFV600E mutation, and ETS1 overexpression—some limitations in sample size and lack of functional validation reduce the immediate translational impact. Nevertheless, the work contributes novel insight into combinatorial biomarker analysis in thyroid cancer.
While the manuscript attempts to address a clinically relevant and timely topic—molecular stratification of PTC through combined analysis of p16 methylation, BRAFV600E mutation, and ETS1 expression—the current evidence is entirely observational. The authors did not perform functional studies to validate the mechanistic link between the biomarkers, and the sample size (n=57) is relatively small, especially for stratified subgroup analyses. This significantly limits the strength of the conclusions. Moreover, the lack of external validation or replication in an independent cohort undermines the generalizability of the findings. Therefore, while the concept is novel and promising, the manuscript in its current form falls short of establishing a robust or clinically translatable molecular model.
Nevertheless, the correlation between p16 methylation and ETS1 overexpression in BRAFV600E-positive tumors is noteworthy and could serve as a starting point for future, more comprehensive studies.
The methodological design includes appropriate molecular techniques: MSP for p16 methylation, MASA-PCR for BRAFV600E detection, qPCR and Western blot/IHC for ETS1 quantification. These are standard, validated methods for analyzing epigenetic and gene expression changes in tumor tissue.
However, the small sample size (n=57) severely limits the statistical power of subgroup analyses—especially when stratifying for combinations of methylation, mutation, and expression status. For example, the group with both p16 methylation and BRAFV600E mutation included only 11 cases, which raises concerns about overinterpretation of statistically significant associations.
Furthermore, although the authors appropriately used non-parametric tests due to the non-normal distribution of data, they did not correct for multiple testing (e.g., Bonferroni or FDR), which increases the risk of Type I error.
Finally, no validation cohort or functional experiments were included, and the study remains purely correlative. The absence of mechanistic data—such as gene silencing or pathway inhibition experiments—weakens the causative interpretation of the observed associations.
The authors interpret the observed correlation between p16 methylation, BRAFV600E mutation, and elevated ETS1 expression as potentially synergistic and prognostically relevant. While the hypothesis is intriguing, the data remain correlative and do not establish causation. The lack of functional assays (e.g., ETS1 knockdown in BRAFV600E-positive cell lines, methylation reversal studies) prevents any mechanistic conclusions.
Clinically, the combination of p16 methylation and high ETS1 expression was associated with higher pT and pTNM stages, suggesting a possible role in risk stratification. However, the small sample size, single-center design, and absence of longitudinal outcome data (e.g., recurrence, survival) limit the immediate utility of this marker panel in clinical decision-making.
The manuscript would benefit from a more cautious interpretation of these results. Statements suggesting that these markers “may guide therapeutic strategies” or “serve as targets” are speculative in the absence of functional validation or drug sensitivity data.
The manuscript is generally well-structured, with clearly defined sections and appropriate figure legends. However, the text would benefit from a more concise and cohesive discussion. Some redundancies and speculative language should be eliminated or toned down. Minor grammatical and syntactical corrections are also needed throughout the manuscript.
Figures are of acceptable quality, though Figure 1 is somewhat complex and would benefit from clearer labeling. Supplementary tables are appropriate and necessary for transparency.
One methodological concern is the inconsistent number of cases analyzed by each technique used to evaluate ETS1 expression (qPCR: n=41, Western blot: n=30, IHC: n=54). The rationale behind this discrepancy is not clearly explained in the manuscript. It would be helpful for the authors to clarify whether this variation was due to sample availability, RNA/protein quality, or a deliberate analytical decision. Such transparency is important to assess potential selection bias and the generalizability of the ETS1-related findings.
Recommendation:
Given the observational nature of the study, the small sample size, and the absence of functional/mechanistic validation, I believe the manuscript requires major revision before it can be considered for publication. I encourage the authors to:
• Clarify the exploratory nature of their findings,
• Temper the clinical claims,
• Expand the discussion on limitations,
• And, if possible, support key findings with additional validation or clearly propose such work in future directions.
See below.
Author Response
Dear reviewer,
Thank you very much for your deep evaluation, your compliments and constructive recommendations concerning our manuscript entitled “Implication of p16 Promoter Methylation, the BRAFV600E Mutation, and ETS1 Expression Determination on Papillary Thyroid Carcinoma Prognosis and High-Risk Patients’ Selection”.
We deeply understand your concerns about the validation of our results and sincerely appreciate your remark regarding the need for functional proof of our findings in a specific experimental model, as conducting such an experiment would significantly enhance our results. We regret to say that we are unable to conduct such additional research due to the lack of the necessary experimental animal models and consumables in our laboratory currently. Unfortunately, it would take a long time to purchase all components and complete all the necessary tasks (in Serbia, it could take more than a year), whereas paper correction takes only 10 days. We believe this would be a valuable next step and hope to explore it in future work.
Furthermore, we have even tried to perform the TCGA analysis, so the results obtained in the present study could be validated and compared with the TCGA dataset. To conduct an integrative biostatistical analysis, we searched for suitable thyroid cancer datasets on the cBioPortal platform (https://www.cbioportal.org/), a publicly available source for genomics, transcriptomics, and methylation data across various tumor types. We identified four papillary thyroid cancer datasets from The Cancer Genome Atlas (TCGA) that we explored in detail.
The first dataset, the TCGA GDC, includes 515 samples and has appropriate clinico-pathological parameters, but lacks methylation data. The second, TCGA Cell 2014, comprises 496 samples and provides suitable clinical parameters for our validation, as well as mRNA expression data for the ETS1 and p16 genes. However, crucial molecular parameters for our comparative analysis — specifically, p16 methylation and ETS1 protein expression data — are unavailable. The third dataset, TCGA Firehose Legacy, includes 399 papillary thyroid cancer samples with detailed clinical information, yet it also does not contain p16 methylation or ETS1 protein expression data. Lastly, the TCGA thyroid carcinoma dataset, which is part of the Pan Cancer Atlas and includes 500 papillary thyroid cancer samples, lacks suitable clinical data for validating our results and does not provide methylation data for the p16 gene.
Additionally, we explored 11 large-scale Pan Cancer datasets for relevant molecular and clinical parameters to corroborate our findings. However, many of these studies featured small cohorts of thyroid cancer samples and did not determine the ETS1 and p16 molecular profiles.
Therefore, unfortunately, we were unable to adequately validate our results with the available TCGA datasets. Even when using only transcriptomic data for the ETS1 and p16 genes, extrapolating these results could lead to misinterpretation, as it is well established that mRNA expression levels do not always correlate with the DNA methylation status of the same gene [1–4].
On the other hand, we have introduced some changes in the manuscript. Before all, we tried to clarify the exploratory nature of our findings and tempered the clinical claims (highlighted in blue, lines: 36, 312,313, 382-384, 423-424). We expanded the discussion on limitations (highlighted in green, lines 398-415) and clearly propose additional validation in future (highlighted in green, lines 410-415). Furthermore, the cause of divergence in sample numbers is explained (highlighted in gray, lines 89-92 and 198-215) and Figure 1 has been modified due to the clearer presentation.
We sincerely hope that all the corrections implemented will meet your requirements and lead to approval.
- Baribault, C.; Ehrlich, K.C.; Ponnaluri, V.K.C.; Pradhan, S.; Lacey, M.; Ehrlich, M. Hypermethylation of Human DNA: Fine-Tuning Transcription Associated with Development 2017.
- Stefansson, O.A.; Sigurpalsdottir, B.D.; Rognvaldsson, S.; Halldorsson, G.H.; Juliusson, K.; Sveinbjornsson, G.; Gunnarsson, B.; Beyter, D.; Jonsson, H.; Gudjonsson, S.A.; et al. The Correlation between CpG Methylation and Gene Expression Is Driven by Sequence Variants. Nat Genet 2024, 56, 1624–1631, doi:10.1038/s41588-024-01851-2.
- Kennedy, E.M.; Goehring, G.N.; Nichols, M.H.; Robins, C.; Mehta, D.; Klengel, T.; Eskin, E.; Smith, A.K.; Conneely, K.N. An Integrated -Omics Analysis of the Epigenetic Landscape of Gene Expression in Human Blood Cells. BMC Genomics 2018, 19, doi:10.1186/s12864-018-4842-3.
- Dhar, G.A.; Saha, S.; Mitra, P.; Nag Chaudhuri, R. DNA Methylation and Regulation of Gene Expression: Guardian of Our Health. Nucleus 2021, 64, 259–270, doi:10.1007/s13237-021-00367-y.
Reviewer 3 Report
Comments and Suggestions for Authors
The authors found that the combination of p16 methylation and high ETS1 protein levels was significantly associated with advanced pT and TNM stages of Papillary thyroid carcinoma. I have the following questions and suggestions:
1.The authors have a limited sample size and lack validation set data, which reduces the credibility of the results. It is recommended to enhance validation by extracting p16 methylation and ETS1 expression data from The Cancer Genome Atlas (TCGA) database or searching for relevant data in the Gene Expression Omnibus (GEO) for validation.
2.In Table 2, chi-square test statistical analysis was used to evaluate the association between methylation of specific CpG regions in the p16 promoter and the occurrence of BRAFV600E mutation in PTC patients, and no correlation was observed. Additionally, in Supplementary Table 1, p16 methylation was not associated with clinical variables such as gender, age, and tumor size. Is the BRAFV600E mutation related to any of these clinical characteristics?
3.The authors reported that the combination of p16 promoter methylation, BRAFV600E mutation, and ETS1 expression level is correlated. Is this combination associated with the patient’s TNM stage?
Author Response
Dear Reviewer,
We would like to thank you for your constructive comments concerning our manuscript entitled “Implication of p16 Promoter Methylation, the BRAFV600E Mutation, and ETS1 Expression Determination on Papillary Thyroid Carcinoma Prognosis and High-Risk Patients’ Selection”. The following are our responses to your comments:
- Thank you for the suggestion as such a comparison would significantly improve our findings. We agree with your remarks; therefore we have tried to validate our findings by TCGA analysis. To conduct an integrative biostatistical analysis, we searched for suitable thyroid cancer datasets on the cBioPortal platform (https://www.cbioportal.org/), a publicly available source for genomics, transcriptomics, and methylation data across various tumor types. We identified four papillary thyroid cancer datasets from The Cancer Genome Atlas (TCGA) that we explored in detail.
The first dataset, the TCGA GDC, includes 515 samples and has appropriate clinico-pathological parameters, but lacks methylation data. The second, TCGA Cell 2014, comprises 496 samples and provides suitable clinical parameters for our validation, as well as mRNA expression data for the ETS1 and p16 genes. However, crucial molecular parameters for our comparative analysis — specifically, p16 methylation and ETS1 protein expression data — are unavailable. The third dataset, TCGA Firehose Legacy, includes 399 papillary thyroid cancer samples with detailed clinical information, yet it also does not contain p16 methylation or ETS1 protein expression data. Lastly, the TCGA thyroid carcinoma dataset, which is part of the Pan Cancer Atlas and includes 500 papillary thyroid cancer samples, lacks suitable clinical data for validating our results and does not provide methylation data for the p16 gene.
Additionally, we explored 11 large-scale Pan Cancer datasets for relevant molecular and clinical parameters to corroborate our findings. However, many of these studies featured small cohorts of thyroid cancer samples and did not determine the ETS1 and p16 molecular profiles.
In conclusion, unfortunately, we were unable to adequately validate our results, so we decided not to present them in the manuscript. Even when using only transcriptomic data for the ETS1 and p16 genes, extrapolating these results could lead to misinterpretation, as it is well established that mRNA expression levels do not always correlate with the DNA methylation status of the same gene [1–4].
-
- Baribault, C.; Ehrlich, K.C.; Ponnaluri, V.K.C.; Pradhan, S.; Lacey, M.; Ehrlich, M. Hypermethylation of Human DNA: Fine-Tuning Transcription Associated with Development 2017.
- Stefansson, O.A.; Sigurpalsdottir, B.D.; Rognvaldsson, S.; Halldorsson, G.H.; Juliusson, K.; Sveinbjornsson, G.; Gunnarsson, B.; Beyter, D.; Jonsson, H.; Gudjonsson, S.A.; et al. The Correlation between CpG Methylation and Gene Expression Is Driven by Sequence Variants. Nat Genet 2024, 56, 1624–1631, doi:10.1038/s41588-024-01851-2.
- Kennedy, E.M.; Goehring, G.N.; Nichols, M.H.; Robins, C.; Mehta, D.; Klengel, T.; Eskin, E.; Smith, A.K.; Conneely, K.N. An Integrated -Omics Analysis of the Epigenetic Landscape of Gene Expression in Human Blood Cells. BMC Genomics 2018, 19, doi:10.1186/s12864-018-4842-3.
- Dhar, G.A.; Saha, S.; Mitra, P.; Nag Chaudhuri, R. DNA Methylation and Regulation of Gene Expression: Guardian of Our Health. Nucleus 2021, 64, 259–270, doi:10.1007/s13237-021-00367-y.
- Thank you for this notice. We have tested the correlation of the BRAFV600E mutation presence with the patients’ clinical characteristics, such as gender, age, and tumor size, and found no significant results (all p > 0.05). You may find the test details in the following table:
|
PTC sample |
gender |
age |
size |
pT |
pTNM |
DTI |
Ei |
ID |
LNM |
|
correlation coefficient |
-0.1954 |
0.0973 |
-0.0825 |
-0.0560 |
0.0987 |
0.0493 |
0.0304 |
-0.1306 |
-0.0308 |
|
p-value |
0.0804 |
0.3875 |
0.4638 |
0.6192 |
0.3805 |
0.6618 |
0.7877 |
0.2452 |
0.7846 |
|
Ei – extrathyroid invasion, ID – intraglandular dissemination, LNM – lymph node metastasis, DTI- degree of tumor infiltration, pT - pT grade, pTNM – pTNM stage (for details se the Material and Methods section). r – correlation coefficient, p-value – statistical significance. Statistical analysis: Spearman’s correlation. PTC: papillary thyroid carcinoma |
|||||||||
Therefore, there was no correlation between the BRAFV600E mutation presence and the tested clinicopathological parameters occurrence (all p > 0.05, tested by Spearman's rank correlation) in tested PTC group of patients. Here, we would like to add, that while the somatic BRAFV600E mutation is an effective diagnostic tool in the context of PTC, its prognostic significance, relationship with enhanced tumor aggressiveness, and poor prognosis in PTC are still being debated. Although some researchers have reported significant associations between BRAFV600E and high-risk clinical-pathological characteristics of PTCs, many others have demonstrated that the BRAFV600E mutation has no significant influence on the poor PTC prognosis in overall analyses. Shortly, recurrence is the most commonly described clinicopathological feature associated with this mutation. However, as the presence of the mutation varies from 30% to 80% of all PTCs, and since the mutation is closely linked with histological features of aggressiveness (such as lymph node metastasis or PTC variant), it is difficult to determine the proportion of risk that is attributable to the BRAFV600E mutation per se or to the other histological changes. In other words, according to previously published data, the presence of the BRAFV600E mutation alone does not appear to be sufficient for risk stratification of PTC patients. And this aligns with the findings of this study. As a result, in this study, we focused on specific additional factors that may operate synergistically to promote PTC aggressive behavior development with the BRAFV600E. Furthermore, as the other reviewer suggested, we have omitted Supplementary Table 1 from the revised version of the manuscript.
3. Thank you for pointing this question out, as it is really an intriguing one. Now we have tested if the p16 promoter methylation, BRAFV600E presence, and ETS1 high expression are associated with the patients’ pT and pTNM stage, and the results are presented in the following table:
|
Clinicopathological parameter |
p16 methylated, BRAFV600E mutated and ETS1 protein levels high (n) |
p-value |
|||||
|
None of three positives |
One of three positive |
Two of three positive |
All three positive |
total |
|||
|
pT |
T1 |
0 |
1 |
0 |
1 |
2 |
0.067 |
|
T2 |
0 |
4 |
6 |
3 |
13 |
||
|
T3 |
0 |
10 |
0 |
3 |
13 |
||
|
T4 |
0 |
0 |
1 |
1 |
2 |
||
|
pTNM |
I |
0 |
7 |
2 |
3 |
12 |
0.129 |
|
II |
0 |
1 |
4 |
1 |
6 |
||
|
III |
0 |
6 |
0 |
3 |
9 |
||
|
IV |
0 |
1 |
1 |
1 |
3 |
||
As could be seen from the table, there are some tendencies that PTC cases with all three tested parameters positive (p16 is methylated, BRAFV600E is mutated, and ETS1 protein levels are high) would probably tend to the higher pT and pTNM stages, but this result is not significant (p>0.05, tested by χ2-test). It's possible that assessing all three parameters is statistically redundant. Testing all three parameters to determine pT or pTNM status may affect the dispersion of the obtained data, resulting in decreased significance. Particularly as shown in the previous answer, BRAVF600E does not correlate with any of the clinicopathological factors in the tested PTC cohort. On the other hand, it reduced the total number of PTC patients included in the test (from 54 to 30). If we could have a larger cohort, perhaps the results would be statistically significant. Furthermore, as we have previously demonstrated, determining just p16 methylation and ETS1 protein levels is sufficient for assessing pT or pTNM status. As a result, from a clinical standpoint, assessing the third element only to determine pT or pTNM status would only increase the cost of the test and the time required to complete it. However, from a biological standpoint, understanding how all tested factors interact is critical to better understanding how the tumor's aggressiveness develops.
According to all the points mentioned above, we introduced some changes in section 3.5 of the manuscript (highlighted in yellow, lines 296-299) and in the discussion section (highlighted in yellow, lines 387-390).
Round 2
Reviewer 2 Report
Comments and Suggestions for Authors
I appreciate the authors' response and their transparency in stating that financial and time constraints prevent them from conducting the additional tests and validations I previously suggested.
While these limitations restrict the study’s comprehensiveness, I recognize the potential scientific value of the topic and consider the findings to be a useful preliminary contribution to the field.
Therefore, I suggest that the authors frame their work more clearly as a pilot or exploratory study. This could include a brief acknowledgment of the methodological limitations in the Discussion section, along with a note encouraging further validation in future research.
With these minor revisions and clarifications, I believe the manuscript could be acceptable in its current form.
Author Response
Dear Reviewer,
The proposed corrections have been incorporated into the manuscript's restriction section (lines 398-401, marked in green). Limitations of the study are clearly presented in lines 393 -398, while lines 401-415 encourage further validation in future research. We genuinely hope that we have now managed to respond to your issues and that our paper is sufficiently improved so it could be published.
Reviewer 3 Report
Comments and Suggestions for Authors
Thank you for addressing my previous inquiries. Additionally, I noted that the immunohistochemistry (IHC) results are solely presented with representative images, lacking semi-quantitative data support. To enhance objectivity and rigor, I suggest supplementing these findings with automated scoring of ETS1 staining intensity using ImageJ's IHC Profiler plugin or comparable software tools. This approach would provide standardized, reproducible quantification of protein expression levels across samples.
Author Response
Dear Reviewer,
The expression of ETS1 in PTC was described and explained in detail in our previous publication (https://doi.org/10.3390/ijms26031253 also cited in this paper). Therefore, we believe that providing the table with the distribution of ETS1 staining in PTC in this study would be similar to presenting previously published data (please see Figures 3 b, f, and h, as well as Figure 4b in the aforementioned paper https://doi.org/10.3390/ijms26031253 ). Additionally, it would contribute to the dispersion of the results presented in this paper, which focuses on the combination of factors that may influence PTC aggressiveness. Furthermore, it might bring to the confusion of the readers, reducing the readability of the paper without adding considerable value to the conclusions. However, we sincerely understand your concerns about the objectivity and reliability of the data presented in this paper; thus, here is a more detailed explanation of the ETS1 immunohistochemical scoring method and the results of ETS1 IHC expression distribution in PTC crossed with p16 promoter methylation.
ETS1 protein staining was performed by two independent researchers, and the results were determined as the average of both researchers’ scores. The staining was evaluated by determining the staining intensity with the following scores: 1—weak intensity, 2—moderate intensity, and 3—strong intensity, as well as by determining the percentage of stained tumor cells (0%–100%). Since the presence of ETS1 was observed in both the nucleus and the cytoplasm of cells, the scoring results were determined separately for the cellular compartments. The staining score for each compartment was calculated as the product of the staining intensity of that compartment and the percentage of stained cells/nuclei, while the overall staining score for each sample was calculated as the sum of the staining scores for both cell compartments (nucleus + cytoplasm). Therefore, the range of staining scores for each compartment ranged from 0 to 3, and consequently, the total IHC score ranged from 0 to 6. A summary of ETS1 IHC staining in PTC is presented in the following table:
Distribution of ETS1 immunohistochemical staining scores in Papillary thyroid carcinoma depending on the p16 methylation status
|
PTC sample |
ETS1 IHC score (n) |
Total (n) |
||||||
|
0-1 |
> 1 - 2 |
> 2-3 |
> 3 - 4 |
> 4 - 5 |
> 5 - 6 |
|||
|
p16 methylated |
no |
6 |
1 |
3 |
9 |
1 |
2 |
22 |
|
yes |
8 |
6 |
8 |
8 |
1 |
1 |
32 |
|
|
Total (n) |
14 |
7 |
11 |
17 |
2 |
3 |
54 |
|
|
PTC: papillary thyroid carcinoma, n: number of cases |
||||||||